# Triptonodiol, a Diterpenoid Extracted from *Tripterygium wilfordii*, Inhibits the Migration and Invasion of Non-Small-Cell Lung Cancer

**DOI:** 10.3390/molecules28124708

**Published:** 2023-06-12

**Authors:** Xiaochen Ni, Xiaomin Jiang, Shilong Yu, Feng Wu, Jun Zhou, Defang Mao, Haibo Wang, Yanqing Liu, Feng Jin

**Affiliations:** 1Department of Respiratory Medicine, The Affiliated Hospital of Yangzhou University, Yangzhou University, Yangzhou 225001, China; nxc19970103@163.com (X.N.); yzwufeng888@163.com (F.W.); zj662121@163.com (J.Z.); 2Institute of Translational Medicine, Medical College, Yangzhou University, Yangzhou 225001, China; 15050735732@139.com (X.J.); yushilong1992@163.com (S.Y.); 13813190949@163.com (D.M.); wanghaibo@yzu.edu.cn (H.W.); yzumpi@163.com (Y.L.); 3The Key Laboratory of Syndrome Differentiation and Treatment of Gastric Cancer of the State Administration of Traditional Chinese Medicine, Yangzhou 225001, China; 4Yangzhou Hospital of Traditional Chinese Medicine, Yangzhou 225001, China

**Keywords:** Triptonodiol, *Tripterygium wilfordii*, non-small-cell lung cancer, invasion and metastasis, autophagy, actin cytoskeleton remodeling

## Abstract

Lung cancer is the most prevalent oncological disease worldwide, with non-small-cell lung cancer accounting for approximately 85% of lung cancer cases. *Tripterygium wilfordii* is a traditional Chinese herb that is widely used to treat rheumatism, pain, inflammation, tumors, and other diseases. In this study, we found that Triptonodiol extracted from *Tripterygium wilfordii* inhibited the migration and invasion of non-small-cell lung cancer and inhibited cytoskeletal remodeling, which has not been previously reported. Triptonodiol significantly inhibited the motility activity of NSCLC at low toxic concentrations and suppressed the migration and invasion of NSCLC. These results can be confirmed by wound healing, cell trajectory tracking, and Transwell assays. We found that cytoskeletal remodeling was inhibited in Triptonodiol-treated NSCLC, as evidenced by the reduced aggregation of actin and altered pseudopod morphology. Additionally, this study found that Triptonodiol induced an increase in complete autophagic flux in NSCLC. This study suggests that Triptonodiol reduces the aggressive phenotype of NSCLC by inhibiting cytoskeletal remodeling and is a promising anti-tumor compound.

## 1. Introduction

According to the WHO, lung cancer is the most common cancer worldwide and the leading cause of cancer deaths [1]. Non-small-cell lung cancer accounts for about 85% of all lung cancer cases. With concerted global efforts, especially in the intensive research on molecularly targeted drugs, the treatment of lung cancer has improved significantly [2]. The latest data show that the 5-year survival rate for lung cancer has exceeded 20% for the first time [1], which is something to be happy about but is still far from satisfactory. In the course of tumor treatment, the occurrence of distant metastasis usually leads to treatment failure, directly causes a poor prognosis, and is the most important cause of death for tumor patients (90%) [3]. Although targets such as human epidermal growth factor receptor-2 (HER2), epidermal growth factor receptor (EGFR), and anaplastic lymphoma kinase (ALK) have been successfully used in the clinic [2], tumors eventually progress due to frequent drug resistance events and the inability of patients to tolerate the toxic side effects of the drugs. Therefore, the development of new drugs for targeting tumor metastasis is now urgently needed.

The detachment of tumor cells from the primary location to distant sites is a very complex process in which the remodeling of actin filaments is necessary and is the mechanical basis for cell movement [4]. In solid tumors, as the tumor grows and the extracellular matrix is extruded, the mechanical forces within it increase, eventually transmitting mechanical signals to the cell interior [5]. In response to these signals, the cytoskeleton will undergo more intense remodeling, conferring greater invasiveness to the cells by altering their mechanical properties and membrane structural fluidity, ultimately causing accelerated metastasis and the progression of the tumor [6]. Therefore, the direct targeting of cytoskeletal remodeling for tumor therapy is a highly effective strategy. Autophagy is a tightly regulated degradation program of the lysosomal pathway that allows cells to engulf a portion of the cytoplasmic contents and plays an important role in cell survival and homeostasis. It has been increasingly shown that natural compounds are among the most common triggers of cellular autophagy. It has been shown that autophagy can inhibit tumor invasion by suppressing tumor cell necrosis and inflammatory infiltration [7]. However, it has also been shown that autophagy increases tumor viability in therapy by scavenging excess ROS or damaged organelles as a means of isolating some of the cellular damage, particularly by reducing apoptosis during distant metastasis [8]. Overall, the available evidence supports the novel notion that early autophagy may be a protective mechanism triggered by a loss in some cellular functions in the face of environmental stress, but that a prolonged and excessive flux of autophagy can cause irreversible damage and ultimately cell death [9].

*Tripterygium wilfordii* Hook F has a long history of use in China for the treatment of tumor-related diseases, first detailed in the Qing Dynasty book “Supplement to Compendium of Materia Medica”. *Tripterygium wilfordii* is found growing in 1219 areas worldwide and is a dominant resource with large reserves [10]. Nowadays, the involvement of traditional Chinese medicine in the clinical treatment of tumors has become a distinctive Chinese therapeutic strategy and has demonstrated remarkable efficacy [11,12,13]. Triptonodiol, a compound in *Tripterygium wilfordii* [14], has been shown to have anti-cancer effects [15]. However, its biological and molecular mechanisms are not yet clear. In this study, we found that Triptonodiol significantly inhibited the migratory and invasive capacity of NSCLC at low cytotoxic concentrations and suppressed cytoskeletal remodeling. At the same time, we also found that Triptonodiol induced an increase in the flux of complete autophagy in NSCLC. We therefore decided to systematically conduct this study to investigate the specific effects of Triptonodiol on the migratory and invasive phenotypes of NSCLC. In many studies of natural compounds, in vitro studies targeting tumor invasion and metastasis are often confounded with many other factors, such as apoptosis, because the natural compounds themselves are tumor-killing [16]. Triptonodiol is therefore also an excellent natural compound for the study of cell motility. The aim of this study is to investigate the specific mechanism of Triptonodiol’s anti-cancer activity, to provide more evidence for the application of *Tripterygium wilfordii*, and to provide more promising lead compounds for the development of new anti-tumor drugs.

## 2. Results

### 2.1. Triptonodiol Inhibits the Migration Ability of NSCLC at Low Cytotoxic Concentrations

Triptonodiol (CAS no. 117456-87-8) is an aromatic abietane diterpenoid compound extracted from *Tripterygium wilfordii*, and its molecular structure is shown in Figure 1B. To initially confirm the inhibitory effect of Triptonodiol on NSCLC, CCK-8 assays were carried out. We treated NSCLC using different concentrations, and the results showed that Triptonodiol was almost non-cytotoxic to NSCLC in the range of 0 μM to 80 μM (Figure 1A). High-content imaging was used to track cell trajectories in real time, and we found that the motility of NSCLC was consistently reduced after Triptonodiol (80 μM) treatment, mainly in the form of decreased mean square displacement and motility velocity (Figure 1C–E). These results indicate that Triptonodiol is able to inhibit NSCLC cell viability and suppress the motility of NSCLC at low toxic concentrations.

### 2.2. Triptonodiol Reduces the Migratory Ability of NSCLC Cells

To observe the degree of migration inhibition on NSCLC by Triptonodiol, wound-healing assays were carried out. We chose three low-toxicity concentrations (20 μM, 40 μM and 80 μM) to study the anti-seeding effects of Triptonodiol. It was shown that the degree of wound healing of NSCLC was significantly reduced after Triptonodiol treatment. These results further confirmed the migration inhibitory effect of Triptonodiol on NSCLC in a dose-dependent manner (Figure 2A,B).

### 2.3. Triptonodiol Significantly Reduces the Invasion Capacity of NSCLC

In living organisms, tumor cells need to lyse the extracellular matrix and invade surrounding tissues in order to detach from the primary focus, a prerequisite for distant metastasis [17]. To investigate whether Triptonodiol has an anti-invasion ability against NSCLC, we conducted Transwell assays. The results showed that Triptonodiol significantly reduced the ability of NSCLC to penetrate the Transwell filter membrane (Figure 3A,B). These results suggest that Triptonodiol can significantly reduce the invasion ability of NSCLC, which is related to the ability of cells to lyse the extracellular matrix.

### 2.4. Triptonodiol Inhibits Cytoskeletal Remodeling in NSCLC

The continuous degradation and remodeling of the cytoskeleton is the mechanical basis for cell motility [4], and we next examined microfilaments and microtubules in NSCLC. After 24 h of Triptonodiol treatment, the actin protein of NSCLC lost its original structure and aggregated around the nucleus, indicating that the remodeling of the cytoskeleton was inhibited. Additionally, the invasive structures of the cell, such as the filamentous and lamellar pseudopods, were reduced (Figure 4A). These are considered to be the hallmark structures for cells to be invasive, especially the filamentous pseudopods. At the same time, we observed no abnormalities in the microtubule structure of the cells. These results suggest that Triptonodiol can inhibit cytoskeletal remodeling by affecting the structure of actin without affecting the structure of tubulin.

### 2.5. Triptonodiol Leads to an Increase in Autophagic Vesicles in NSCLC Cells

There is growing evidence that the drug modulation of cellular autophagy levels is an important cellular response in tumor therapy [18]. Therefore, we investigated whether autophagy levels in NSCLC were regulated by Triptonodiol. Treatment with Triptonodiol increased the number of autophagic vesicles inside the cells (Figure 5A). Immunofluorescence experiments showed that these autophagic vesicles surrounded the nucleus, which may have been a manifestation of complete autophagy [19]. Next, we examined the expression level of the LC3 protein by means of immunoblotting experiments. The results showed that treatment with Triptonodiol on NSCLC significantly increased the expression of the LC3-Ⅱ protein (Figure 5B,C), a marker of mature autophagic vesicles [20].

### 2.6. Triptonodiol Induces Complete Autophagic Flux in NSCLC

Autophagy is a dynamic process whereby autophagic vesicles engulf specific cytoplasmic structures and transport them to lysosomes for degradation. Both the early activation of autophagy and the late inhibition of autophagy exhibit the accumulation of the LC3 protein [20]. Therefore, to investigate how Triptonodiol-induced LC3 accumulation is generated, we labeled the LC3 protein with RFP and GFP. When autophagy enters the final step and fuses with lysosomes, the green fluorescent protein GFP is inactivated by a low pH, but the red fluorescent protein RFP maintains its normal function. It can be seen that the red fluorescence accumulates in Triptonodiol-treated NSCLC cells, while the green fluorescence is quenched (Figure 6A), indicating that these autophagosomes complete fusion with lysosomes. Chloroquine (CQ) is able to block the fusion of autophagosomes and lysosomes and up-regulate the lysosomal pH [9], so that the green fluorescence is not disrupted and serves as a control group for this experiment. As shown, the green fluorescent protein GFP on LC3 could not be quenched by the lysosome after chloroquine treatment. These results suggest that Triptonodiol induces autophagic activation and increases the complete autophagic flux in NSCLC.

## 3. Discussion

According to the American Cancer Society, 230,000 new cases of lung cancer are predicted to be diagnosed in 2023, and this number will continue to increase [1]. When lung cancer is confined to the primary location, patients will have the opportunity to undergo radical surgery or adjuvant chemotherapy followed by surgery. Typically, when lung cancer has metastasized distantly, the prognosis for patients is often poor [21]. With the development of molecular biology, a large number of biomarkers have been identified that play an important role in tumor progression [22]. Molecularly targeted drugs based on this have now achieved great success in clinical treatment due to their extremely high bioactivity and targeting properties [23]. However, due to the frequent occurrence of drug resistance events in lung cancer, the molecularly targeted drugs available on the market are still far from meeting the needs of patients [24]. Therefore, the search for and development of new drugs for oncology treatment remains a major task at present. Triptonodiol, a compound found in *Tripterygium wilfordii*, was found to significantly inhibit the spreading ability of NSCLC and induce cytoskeletal remodeling inhibition, as well as autophagy activation. Our study confirms that Triptonodiol is a potent inhibitor targeting NSCLC invasion and migration, and preliminarily explores its possible mechanisms.

Through wound-healing assays, Transwell penetration assays, and real-time dynamic tracking techniques of cell trajectories, we confirmed that Triptonodiol reduces the invasion and metastasis abilities of NSCLC. We found that Triptonodiol inhibited cytoskeleton remodeling and disrupted actin arrangement. Actin-enriched cell membrane projections are a key feature of aggressive tumor cells. Cell motility requires intense cytoskeletal remodeling, a dynamic process in which the cytoskeleton undergoes continuous degradation and remodeling, thereby driving continuous cell displacement [4,7]. What can be confirmed is that the invasion and metastasis of a wide range of human diseases, including cancer cells, are associated with abnormal and dysregulated cell motility and invasion. The acquisition of a highly motile and invasive phenotype is characteristic of aggressive tumor cells and is necessary for the cells to invade vascular or lymphatic structures [25]. Lamellipodia is a raised structure of the cell membrane at the front of the cell that propels the cell membrane progressively forward by polymerizing actin to generate sufficient mechanical force [26]. Filopodia, normally a marker of tumor cell aggressiveness, has been found to penetrate the lung parenchyma in mouse breast cancer [27] and has been shown to increase progressively with tumor progression [26]. We found that actin in Triptonodiol-treated NSCLC cells was dispersed in the cytoplasm and no longer aggregated into filamentous actin polymers. At the same time, Triptonodiol reduced the formation of Lamellipodia and Filopodia. While studies on the various components of the cytoskeleton have typically been independent, there is growing evidence that microtubules (MTs) and microfilaments (actin) are directly mutually supportive [4]. Therefore, we also studied the structural changes in MTs. It was not observed whether Triptonodiol was able to alter the structure of microfilaments, but this is consistent with our findings on autophagy, which we will discuss below. We therefore suggest that a Triptonodiol-mediated reduction in the migratory and invasive capacity of NSCLC is caused by cytoskeletal remodeling.

Autophagy is a very ancient cytoplasmic program that allows cells to “eat” parts of themselves [9]. The biological significance of autophagy for tumor cells is largely dichotomous, promoting tumor progression or inhibiting it [28]. For example, excessive autophagy causes autophagic death [29], while protective autophagy is initiated in response to excess ROS or damaged organelles produced by drugs [30]. However, for tumor invasion and metastasis, autophagy usually shows a detrimental effect because it is a way of gaining more viability and resilience by “digesting” itself at the cost of giving up some cellular functions [31]. Our study found that Triptonodiol significantly enhanced autophagy levels in NSCLC. At the same time, we found that this form of autophagy is complete autophagy, in which NSCLC cells finalize the recycling and reuse of intracellular materials by engulfing some cellular structures and transporting them to lysosomes [20]. We usually refer to this complete process as autophagic flux. The activation of autophagy may be one explanation for the inhibition of NSCLC migration and invasion by Triptonodiol. Current evidence suggests that once the autophagic program is initiated, any factor that interrupts its progression will cause damage to the cell, also known as incomplete autophagy [19]. Further modulation of autophagic flux will alter the cytotoxic effects of Triptonodiol on NSCLC, and therefore it is difficult for us to confirm whether a Triptonodiol-induced increase in autophagic flux is a key factor affecting the ability of NSCLC to spread. It has been shown that the fusion of autophagic vesicles with lysosomes requires the involvement of microtubules, which act as delivery tracks for autophagic vesicles and ultimately complete the autophagic flow process [32]. This is consistent with our finding of no abnormal microtubule structure in Triptonodiol-treated NSCLC, otherwise, the autophagic process might not be fully underway. There is a very interesting study that has to be mentioned here. In E-Cadherin-deficient tumor cells, although local invasion was increased, the accumulation of ROS (reactive oxygen species) resulted in a substantial decrease in distant metastatic events [33]. Subsequent studies confirmed that the inhibition of autophagy, although having no effect on cell invasion, metastasis and EMT processes, nevertheless increased anoikis during subsequent tumor metastasis [34]. Thus, autophagy can be a cytoplasmic process that promotes distant metastasis and increases cellular resistance to anoikis by inactivating excess ROS and damaged organelles, which needs to be viewed with caution. Thus, the inhibitory effect of Triptonodiol on NSCLC in vitro may not be the key rate-limiting step for its metastasis in vivo due to the presence of autophagy. At the same time, the cytoskeleton in organisms is also regulated by the surrounding extracellular matrix and regulates the aggressive phenotype of tumor cells in a pattern of combined chemical and mechanical signaling. We will next study the effect of Triptonodiol on distant colonization in vivo.

## 4. Materials and Methods

### 4.1. Reagents and Antibodies

LC3 (catalog number: 3868S) and GAPDH (catalog number: 5174S) antibodies were purchased from Cell Signaling Technology (Boston, MA, USA). Cell Counting Kit-8 was purchased from Beyotime (Shanghai, China). Actin (catalog number: C2203S) and tubulin (catalog number: C1051S) staining reagents were purchased from Beyotime (Shanghai, China). Matrix (catalog number: 356234) was purchased from Corning Incorporated (New York, NY, China). The sequencing and packaging of the virus carrying LC3B (catalog number: NM_022818) was completed by Genechem Co., Ltd. (Shanghai, China). Triptonodiol (B32757) was purchased from Shanghai Yuanye Bio-Technology Co., Ltd. (Shanghai, China).

### 4.2. Cell Culture

H1299 (catalog number: CL-0165) and A549 (catalog number: CL-0016) cells were purchased from Procell Life Science & Technology Co., Ltd. (Wuhan, China). The cell culture conditions were performed strictly following the instructions. H1299 cells were cultured using RPMI 1640 (Cytiva, catalog number: SH30809.01) and A549 cells were cultured using F-12K (Procell, catalog number: PM150910). The culture medium was supplemented with 10% serum and did not contain antibiotics. The parameters of the incubator were set to 5% CO_2_, 37 °C and saturated humidity.

### 4.3. Cytotoxicity Detection

The cytotoxicity of Triptonodiol to NSCLC was assayed via the CCK-8 assay. The procedure for this experiment was carried out in strict accordance with the instructions. Briefly, 10,000 cells per well were grown in 96-well plates, and after the cells had recovered their morphology, NSCLC cells were treated with Triptonodiol for 24 h. The drug-containing medium was aspirated to exclude the effect of the drug on absorbance. Cells were then incubated with 10% CCK-8 solution for 1 h at 37 °C, protected from light, and absorbance values were measured at 450 nm.

### 4.4. Wound-Healing Assay

The motility of the NSCLC cell was assayed using the wound-healing assay. Cells were cultured in 6-well plates, and we waited for cells to grow to 95% confluence. A sterile pipette was used to create a straight wound on the cell layer and the floating cells were washed off. Subsequently, drug-containing serum-free medium was added and the culture was continued for 24 h before being photographed and analyzed [35].

### 4.5. Cell Dynamic Tracking

To detect the persistence of the migration inhibition of NSCLC cells by Triptonodiol, a high-content imaging system was applied for cell trajectory tracking. After treating the NSCLC cells with Triptonodiol for 12 h, they were placed into the imaging system for delayed imaging for 12 h. The parameters for taking pictures were set to take 9 fields of view for 1 well every 10 min. Cell movement trajectories were drawn using a high-content imaging system and data analysis was performed using GraphPad prism software, version 9.5.1.

### 4.6. Transwell Assay

The invasiveness of NSCLC was detected using Transwell chambers. The matrix was spread evenly over the upper chamber and incubated at 37 °C for 2 h. After the matrix had solidified, 20,000 NSCLC cells per well were uniformly inoculated in the upper chamber. The medium in the upper chamber was free of serum and the medium in the lower chamber contained 10% serum and various concentrations of Triptonodiol. Twenty-four hours later, the cells in the upper chamber were wiped off and the cells in the lower chamber were fixed with 4% paraformaldehyde. Subsequently, staining and photography were performed, and cell counting was performed using ImageJ.

### 4.7. Immunofluorescence

Immunofluorescence was used to detect the expression and localization of intracellular proteins. Briefly, after 24 h of Triptonodiol treatment for NSCLC, cells were fixed using 4% paraformaldehyde, washed once with PBS and then permeabilized with 0.3% Triton X-100 for 30 min, followed by 3 washes with PBS. This was followed by a block program at room temperature using 5% BSA for 2 h. Typically, the incubation conditions were 14 h at 4 °C for primary antibodies and 1 h at room temperature for secondary antibodies. DAPI staining was performed for 15 min at room temperature, followed by washing and imaging.

### 4.8. Transfection

The sequence of the virus was designed and packaged by Shanghai Genechem Co., Ltd., and we infected the H1299 cell line and the A549 cell line strictly according to the instructions. Briefly, the cells were adjusted to a low confluency (approximately 20%) and when their condition returned to normal, serum-free medium with virus particles and infection-boosting solution (provided by the company) were added. The incubation time was 16 h, and after which the medium with the virus was removed and washed three times. The culture was continued for about 3 days and the cells were observed using fluorescence microscopy. When bright fluorescence was observed, 2 μg/mL of puromycin was added to the medium. After maintaining the screen for 2 days, the cell line was considered to be successfully constructed when each cell was fluorescent.

### 4.9. Western Blot

Immunoblotting was used to detect the expression of intracellular proteins. Briefly, RIPA buffer was used to lyse the cells, followed by the BCA assay to measure the protein concentration. After mixing the loading buffer and protein lysate in proportion, SDS-PAGE electrophoresis was performed. The proteins on the gel were then transferred to the PVDF membrane by means of horizontal electrophoresis. The PVDF membranes were sealed with 5% skimmed milk powder for 2 h at room temperature and then washed 3 times with TBST for 10 min each. The subsequent steps were performed according to the same principle as other immunoreaction experiments, with incubation conditions of 14 h at 4 °C for primary antibody and 1 h at room temperature for secondary antibody. The relative expression of the proteins was then detected using a gel imaging system.

### 4.10. Statistical Analysis

SPSS 25.0 was used for statistical analysis. Differences between two groups were tested using *t*-tests, and differences between more than three groups were tested using one-way ANOVA. The differences were considered to be statistically significant when * *p* < 0.05, ** *p* < 0.01 and *** *p* < 0.001.

## 5. Conclusions

Overall, our data demonstrated that Triptonodiol inhibits NSCLC migration, which is likely to be associated with cytoskeletal remodeling inhibition and autophagy activation. Triptonodiol significantly inhibits NSCLC invasion and migration at low cytotoxic concentrations in vitro, which is a potential lead compound for tumor therapy.

## Figures and Tables

**Figure 1 molecules-28-04708-f001:**
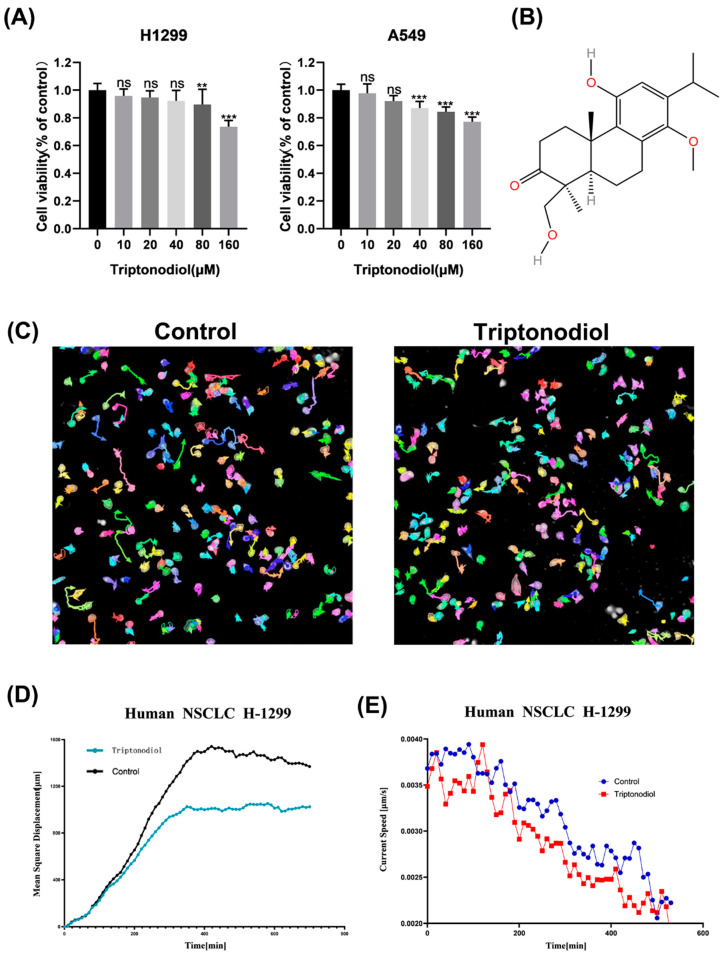
Triptonodiol inhibits the migration ability of NSCLC at low cytotoxic concentrations. (**A**) CCK-8 assay of NSCLC after 24 h of Triptonodiol treatment. (**B**) Chemical structure formula of Triptonodiol. (**C**) Motion trajectory of NSCLC depicted over 12 h. (**D**) Mean square displacement of NSCLC. (**E**) Real-time motion velocity of NSCLC. ns, *p* > 0.05; **, *p* < 0.01; ***, *p* < 0.001 vs. Ctrl (Control).

**Figure 2 molecules-28-04708-f002:**
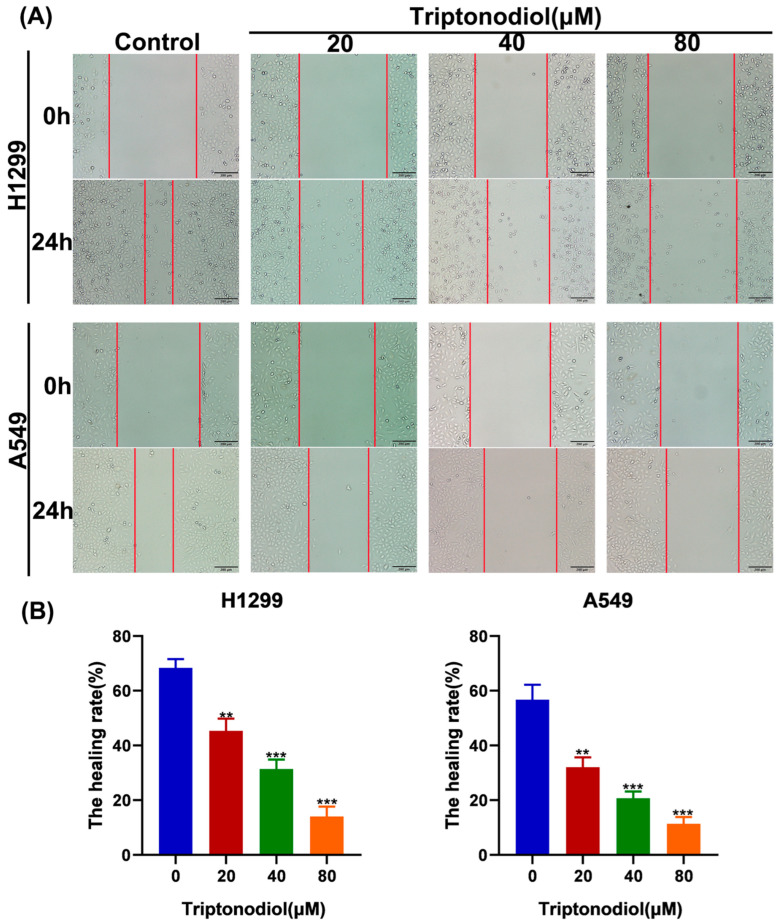
Triptonodiol reduces the migratory ability of NSCLC cells. (**A**) Wound healing of NSCLC cells before and after 24 h of treatment with different concentrations of Triptonodiol. Scale bar, 200 μm. (**B**) Statistical analysis of the wound healing rate of NSCLC cells. **, *p* < 0.01; ***, *p* < 0.001 vs. Ctrl (Control).

**Figure 3 molecules-28-04708-f003:**
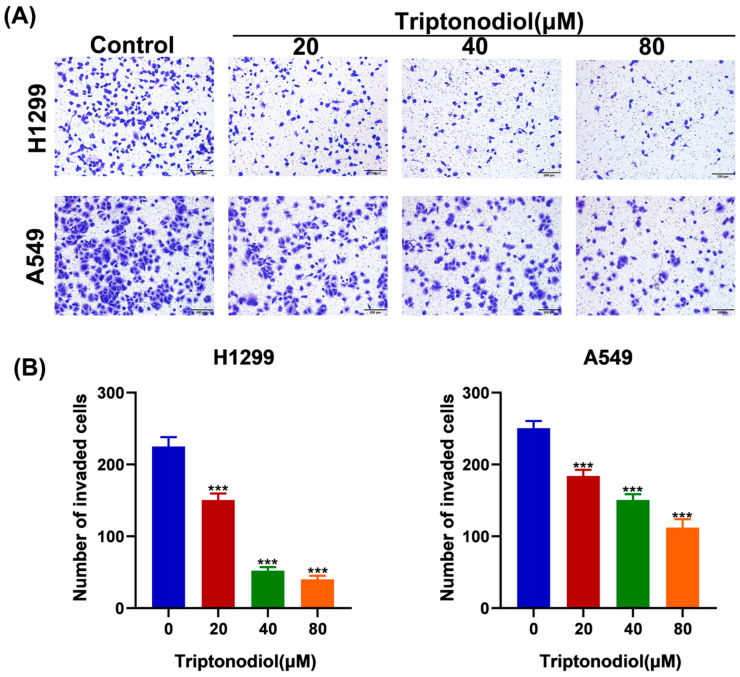
Triptonodiol significantly reduces the invasion capacity of NSCLC. (**A**) Images of cells penetrating the Transwell filter membrane after 24 h of treatment with different concentrations of Triptonodiol. Scale bar, 200 μm. (**B**) Statistical analysis of the number of NSCLC cells that penetrated the filter membrane. ***, *p* < 0.001 vs. Ctrl (Control).

**Figure 4 molecules-28-04708-f004:**
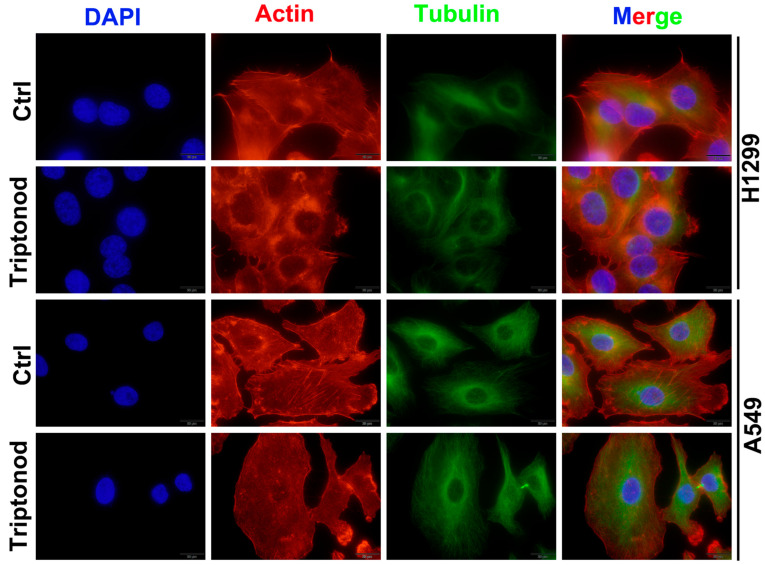
Triptonodiol inhibits cytoskeletal remodeling in NSCLC. Immunofluorescence staining of actin and tubulin after 24 h of Triptonodiol treatment at concentration of 80 μM. DAPI was used for nuclear staining. Scale bar, 20 μm.

**Figure 5 molecules-28-04708-f005:**
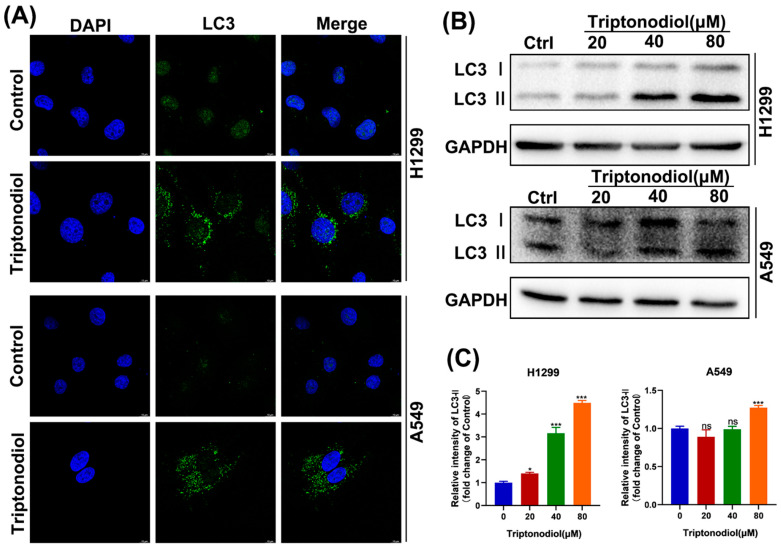
Triptonodiol leads to an increase in autophagic vesicles in NSCLC cells. (**A**) Immunofluorescence staining of LC3 for Triptonodiol (80 μM)-treated NSCLC for 24 h. DAPI was used for nuclear staining. Scale bar, 20 μm. (**B**) Immunoblot of LC3 for different concentrations of Triptonodiol-treated NSCLC for 24 h. GAPDH was used as an internal reference. (**C**) Statistical analysis of immunoblots. ns, *p* > 0.05; *, *p* < 0.05; ***, *p* < 0.001 vs. Ctrl (Control).

**Figure 6 molecules-28-04708-f006:**
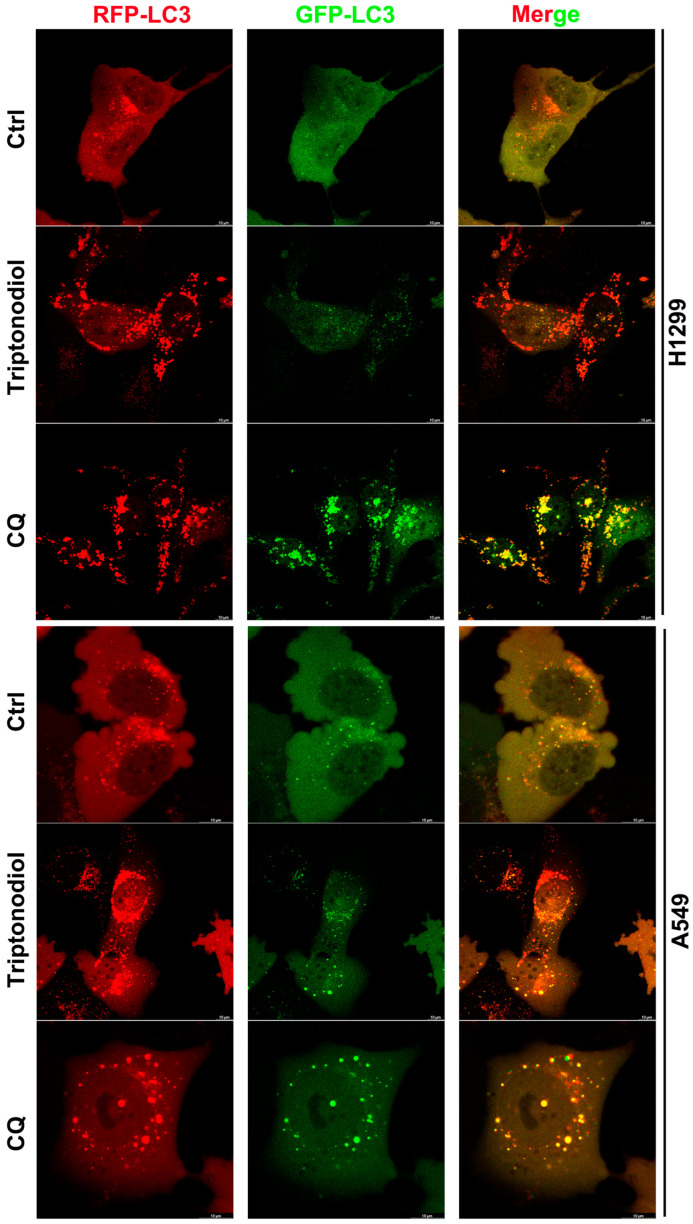
Triptonodiol induces complete autophagic flux in NSCLC. Fluorescence-coupled protein expression was detected in retroviral-infected NSCLC after treatment with Triptonodiol (80 μM) or Chloroquine (20 μM) for 24 h. Scale bar, 10 μm.

## Data Availability

All data were obtained from Yangzhou University and affiliated institutions, and the corresponding author can provide all original content. We declare that no data were derived from third parties, and no paper mill was used.

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
