# Peer review of "Triptonodiol, a Diterpenoid Extracted from Tripterygium wilfordii, Inhibits the Migration and Invasion of Non-Small-Cell Lung Cancer"

_molecules, 2023, doi:10.3390/molecules28124708_

Round 1

Reviewer 1 Report

The importance of the search for new anticancer drugs is difficult to underestimate. The authors present their study of triptonodiol against non-small lung cancer A549 as the pioneer work. Unfortunately, it is not correct. In 2016, a group of Chinese scientists studied the cytotoxic activity of the group of natural diterpinoids including triptonodiol against different cancer cell lines (BMCL, 2016, 26, pp.2942-2946). One of the cell lines was A549. The compound showed micromolar cytotoxicity (IC50=21.59uM). The authors can’t ignore the previous studies of this compound and write that triptonodiol is a novel diterpenoid. They made some interesting findings in the mechanism of the compound’s action, but their ignorence of the literuture data is unacceptable. I have to reject this paper in the present format.

Author Response

We apologise that we are not deliberately ignoring previous research, we wish to show that the mechanism of action of this compound on lung cancer is a novel study. Please forgive our poor English skills for this serious misrepresentation. We have corrected the inappropriate description and invited a professional to make further corrections to the language. In the meantime, we cite the above-mentioned literature and acknowledge this pioneering work. The corrections we have highlighted in red. We are willing to revise again if there is still something inappropriate.

Reviewer 2 Report

The authors present an interesting and original manuscript regarding the mechanism of triptonodiol to inhibit molecular processes that enhance a more aggressive phenotype including migration and invasion through actin remodeling inhibition and autophagy activation in NSCLC cells. This topic is relevant given the ability of NSCLC to generate distant metastases in its early stages, being also able to colonize any human organ.   Methods need to be explained in detail, including catalog number, brand, and country of origin of antibodies and cells.   The result section must be reviewed in detail since the figures do not correspond to the figure caption or to the text of the manuscript, which makes it very difficult to analyze the information. However, by relating the text with the correct corresponding figure, it can be verified that the compound does indeed have an effect on the invasion and migration of NSCLC cells.   English needs to be improved as the author abuse of the past perfect tense to present abstract and methods It would be desirable to have sharper images of western blot   All images must be reviewed for correctness and positioned in place to avoid confusion.

Author Response

We are very sorry for the misplaced images and we have now corrected the order of the images. We have made some additions to the methods so that readers can better understand and reproduce our results. For those antibodies and reagents that may have a significant impact on the results of the experiment, we have labelled them with their brand and Catalogue number. As English is not our native language, we do currently have problems with its use, so we have had the entire manuscript linguistically edited by professionals. Western blot images of H1299 are relatively clear, individual images of A549 are not as clear. If the journal requires clearer images, we will complete these experiments again, which is expected to take around 15 days. The corrections we have highlighted in red. We are willing to revise again if there is still something inappropriate.

Reviewer 3 Report

The author advised for rephrasing the abstract.

The author mentioned that the studied molecule is already patented and used in Chinese medicine. Therefore author advised, to elaborate the discussion with relevant citations.

In result Fig 2A, at 160 micromolar concentrations of Triptonodiol error bars have large difference between both studied cell lines. Please justify the obtained result.

Author Response

We have revised the abstract to state the main results. COE (Celastrus Orbiculatus Thunb extract) is our patented drug, but the individual compound Triptonodiol in it is not our patent. As Triptonodiol was first discovered in Tripterygium wilfordii, we respect and appreciate the previous research and have therefore changed the description of this compound.

We have carefully reviewed the data and found an absorbance value of 0.121 in the data for A549, which was not found in previous data processing work. The value of 0.121 was approximately the background reading for CCK-8, so we considered this data to be an experimental technical error and we removed it. We have reviewed each of the readings and consider all other values to be plausible. Please see the attachment for detailed data.

Reviewer 4 Report

The review of the paper entitled:

“Triptonodiol, a novel diterpenoids extracted from Celastrus orbiculatus Thunb, inhibits the migration and invasion of non-small cell lung cancer”

The manuscript reported the inhibitory activity of triptonodiol on the NSCLC migration and invasion of NSCLC. The authors have found that Triptonodiol significantly inhibited invasion and migration of NSCLC by actin remodeling inhibition and autophagy activation.   

The results were not presented clearly. I recommend the manuscript will be reconsidered after major revisions.

1.      Title: removed the word “novel” since triptonodiol was isolated long time ago.

2.      The latin name Celastrus orbiculatus should be in italic in the entire manuscript.

3.      Rewrite the abstract, in my opinion it’s not necessary to mention all the methods here.

4.      Line 54-55: it’s not correct to mention the ref [8] here.

5.      Line 111: Blot

6.      Line 132: Write clearly about 3 selected concentrations in your study

7.      Figure 2 and Figure 1 should be exchanged.

8.      Figure 2B is missing.

9.      What is the control in the experiments? What is CQ in Figure 1A ?

Where is the figure 5B, 5C

1.      I can not see the file in SI

Author Response

  1. We have removed the word "novel". We intended to convey that this compound was the first to be found to inhibit the migration and invasion of lung cancer in this study, but the misuse of English caused a misunderstanding.2. We have italicised the Latin names of the plants and checked the writing of the Latin names as required (http://www.theplantlist.org/tpl1.1/search?q=Tripterygium+wilfordii). 3. We have rewritten the abstract to focus on the main findings and results.4. We have corrected the description to state that triptonodiol was first isolated from Tripterygium wilfordii. This was a mistake we made when reading the literature and misunderstood and omitted some information, so apologies. 5-8. We have corrected these descriptions and ordered the images correctly. 9. CQ is chloroquine, a late autophagy inhibitor used to block the fusion of autophagosomes and lysosomes. We have now added chloroquine to the Manuscript with the full name and checked the numbering of all images.

Round 2

Reviewer 1 Report

The authors improved the manuscript significantly. Still I would like to see the references for standard procedures they followed for cytotoxicity assay and wound healing assay.

Author Response

We have provided references to the literature on wound healing assay, which are highlighted in yellow. For the cytotoxicity assay, we refer to the standard procedure in the product instructions. The procedure is the same as the one provided by MCE. www.medchemexpress. cn/inhibitor-kit/cell-counting-kit-8.html

Reviewer 2 Report

the manucript has been improved. It can be accepted in it present form

Author Response

Thank you for your review, we have made minor changes to the manuscript, added references to wound healing experiments and highlighted them in yellow.

Reviewer 4 Report

The review of the paper entitled:

Triptonodiol, diterpenoid extracted from Tripterygium wilfordii, inhibits the migration and invasion of non-small-cell lung cancer

The authors have made correction to improve the quality of manuscript. The manuscript presented interesting results. The manuscript can be accepted after minor revision.

Line 75-78: Please consider the sentences. You mentioned about the results of this work, but why you use “recent study” ? Please cite the references.

Author Response

We apologise that this is a problem with our expression, this is a result of this study and not other prior studies. We have amended this sentence and highlighted it in yellow.